# Investigations of the Hydrogen Bonds and Vibrational Spectra of Clathrate Ice XVI

**DOI:** 10.3390/ma12020246

**Published:** 2019-01-12

**Authors:** Ze-Ren Wang, Xu-Liang Zhu, Lu Jiang, Kai Zhang, Hui-Wen Luo, Yue Gu, Peng Zhang

**Affiliations:** School of Space Science and Physics, Shandong University, Weihai 264209, China; wangzeren96@163.com (Z.-R.W.); zhuxuliang@outlook.com (X.-L.Z.); Jiang_Lu@163.com (L.J.); 17862711589@163.com (K.Z.); luohuiwen1900@163.com (H.-W.L.); du_guyue@163.com (Y.G.)

**Keywords:** inelastic neutron scattering, ice XVI, vibration mode, hydrogen bond

## Abstract

Natural gas hydrates are ice-like crystalline materials formed from natural gas and clathrate ice under high pressure and low temperature. Ice XVI, the first S-II type clathrate ice produced in the lab, was simulated by first-principles density functional theory with the CASTEP code. A 34-molecule supercell was built to mimic the hydrogen-disordered structure. The vibrational spectra were calculated as a reference for inelastic neutron scattering (INS), infrared (IR) absorption, and Raman scattering experiments. Two kinds of H-bond vibration modes corresponding to two different bond strengths were found in our previous studies. In this paper, the statistics of distribution calculated by integrating these two kinds of modes was found to match the phonon density of states (PDOS) very well. We confirmed that the two basic types of H-bonds also appeared in clathrate ice XVI. The typical normal modes were analyzed to illustrate the dynamic process of lattice vibrations.

## 1. Introduction

Ice has a diverse presure temperature (P–T) diagram, with currently more than 17 crystal phases known under a variety of pressures and temperatures, including clathrate ice under negative pressure [1,2,3,4,5,6,7,8,9,10,11]. Natural gas hydrates are crystalline compounds composed of clathrate ice and natural gas. Compared with petroleum and coal, natural gas produces less carbon dioxide and pollution. The amount of gas hydrate stored in the earth is twice that of the existing oil and gas reserves and will be an important energy source in the future [12,13,14,15,16]. In the past, it was thought that empty hydrated lattices could not exist because guest molecules were needed to stabilize the host framework [17,18,19]. Later, Wooldridge and Jacobson et al. suggested that these guests could be removed, leaving a metastable, empty inclusion compound [20,21]. In 2014, Falenty et al. experimentally prepared a clathrate ice crystal by evacuating the S-II type cage hydrate [10]. The resulting crystal, named ice XVI, has the lowest density of any member of the ice family found to date (0.81 g/cm^2^) and its discovery opened up research into negative-pressure clathrate hydrates. Ice XVI was later reported to have good mechanical stability and negative thermal expansion [22]. However, information on its vibrational spectrum is lacking in the literature. This work focuses on the phonon statistics of ice XVI and prediction of its IR absorption, Raman scattering, and inelastic neutron scattering (INS) spectra. The typical vibration modes are discussed to clarify the dynamic process of lattice vibrations.

## 2. Computational Methodology

Using the CASTEP code [23], a first-principles density functional theory (DFT) method, we calculated the lattice vibrations of the ice XVI phase. On the basis of our preliminary calculations, the exchange-correlation (XC) function of revised Perdew-Burke-Ernzerhof (RPBE) [24], one of the generalized gradient approximation (GGA) method, was selected for geometry optimization. The energy convergence threshold of the self-consistent field (SCF) for geometric optimization was set as 1.0 × 10^−9^ hartree. The energy cut-off was set as 750 eV and the K-point separation was 0.07/Å. Norm-conserving pseudopotentials were used to calculate the phonon density of states (PDOS). We took the periodic structure of ice XVI from Ref. [10]. The original crystal data only included the positions of oxygen. To obtain a complete structure, a program was designed to position hydrogen atoms randomly within a 34-molecule supercell. The number of possible permutations and combinations of hydrogens is very large. In practice, we reduced the number of possible states by restricting the search to those with a uniformly distributed dipole moment. Thus, the hydrogen atoms were disordered, and the water molecules were linked by hydrogen bonds (H-bonds) to form a clathrate structure as shown in Figure 1. The supercell contained a cage composed of 28 molecules, forming 12 5-membered H-bonded water rings and 4 6-membered rings, denoted 5^12^6^4^. These rings were capable of enclosing large guest molecules. The geometry optimization was performed under zero pressure. Because RPBE slightly underestimates H-bond interactions, the density of the calculated crystal was 0.77 g/cm^2^, 5% lower than the experimental value. The H-bond lengths ranged from 1.797 Å to 1.845 Å, the length of O–H covalent bonds ranged between 0.984 and 0.990 Å, and the distance between adjacent oxygen ranged from 2.768 to 2.83 Å.

In addition, we counted the numbers of four-H-bond and two-H-bond normal modes based on the calculated results using a purpose-designed, self-compiled program. To clearly characterize the complex vibration modes, which are a key property of ice hydrates, we distinguished the mode types by one molecule in the supercell possessing the largest amplitude. The standard classification of stretching modes in terms of the vibrational direction was applied. Finally, we plotted a distribution diagram to mimic the PDOS of these two types of H-bonds.

## 3. Results and Discussion

Based on the harmonic approximation, the PDOS, infrared (IR) absorption, and Raman scattering spectra of ice XVI were calculated as shown in Figure 2. In theory, the PDOS spectrum integrates all phonons throughout the first Brillouin zone (BZ). The INS signals are proportional to the PDOS and can be compared with the PDOS spectrum. However, the interactions between photons and phonons occur only near the center of the BZ. The peaks in the Raman and IR spectra correspond to the frequencies of the calculated normal modes at the gamma point. Thus, we used these three simulated spectra as reference data for the INS, Raman, and IR experiments.

Because of the wide range of spectral intensities of Raman and IR, Figure 2 presents four sections of each of the spectra separately for comparison, on appropriately adjusted intensity scales. For a unit cell containing 34 molecules, there are 34 × 3 × 3 − 3 = 303 optical normal vibration modes. To illustrate the dynamic process of the normal modes, some typical modes of each section are presented for discussion.

The most interesting results are in the translational band. For a water molecule, there are three spatial vibration modes: stretching, wagging, and rocking. The mechanism of H-bond vibrations was the central topic in our previous studies [25,26,27,28,29]. We found two basic H-bond vibration modes in ice Ic, which gave rise to two triangular peaks in INS [27]. In all ice phases, each water molecule is linked with four neighbors via H-bonds to form a tetrahedral structure. According to our investigations, in such a local structure, when the central molecule vibrates along its angle bisector, the four connected H-bonds oscillate simultaneously to yield a four-bond mode. In contrast, there also exist vibrational motions along two H-bonds while the other two remain unchanged, corresponding to a two-bond mode. The vibration modes of H-bonds can thus be classified into two groups, four-bond modes and two-bond modes. We integrated these two groups in the region from 160 cm^−1^ to 320 cm^−1^ to mimic the PDOS spectrum of the H-bonds, as shown in Figure 3. The resulting spectrum matches the H-bond peaks in Figure 2 very well. The two-bond modes are distributed in a lower energy range while the four-bond modes are much stronger. In ideal ice Ic, the strength ratio of these two modes is $\sqrt{2}$. Considering the distance between nearest neighbors in the hydrogen-disordered structure of ice XVI, the lengths of the H-bonds are calculated to range from 1.797 Å to 1.845 Å. Therefore, the strengths of the two kinds of modes have a distribution that gives rise to two main peaks, in which some values overlap. The simulated data confirmed that the two types of H-bonds also occur in clathrate ice, which may be of interest to researchers into natural gas hydrates.

For example, in the normal mode at 314 cm^−1^ shown in Figure 4, the molecules mainly stretch along the bisector of the angle in-plane. The molecule oscillates relative to its four neighbors in this mode. Under this situation, the four H-bonds act in synchrony. However, for the mode at 220 cm^−1^, the molecules undergo a combination of rocking and wagging along the H-bonded linkages. The vibration of each molecule only aligns with two H-bonds. Herein, the strength ratio of these two modes is approximately equal to $\sqrt{2}$. We provide a video that depicts the normal modes at 220 cm^−1^ and 314 cm^−1^ to illustrate the dynamic process (Appendix A). As shown in Figure 2, in the translational band, the Raman spectrum has a distinct peak at 212 cm^−1^. There is also a distinct peak at 175 cm^−1^ in the IR spectrum. Both modes correspond to the weak H-bond vibration and together make up the weak peak at around 200 cm^−1^. The main peak of the PDOS at around 300 cm^−1^ is inactive for IR absorption and very weak for Raman scattering.

For the intermolecular librational band, the frequency range is from 592 cm^−1^ to 1049 cm^−1^ with 102 normal mode patterns. This number is three times the number of molecules within a unit cell because there are three vibration modes for each molecule: rocking, wagging, and twisting. Note that the molecules move through space in the translational band, while the oxygen atoms remain static in the librational band. Very interestingly, one uniform vibration was found in this region for this ice phase. As shown in Figure 4, the absorption at the frequency of 592 cm^−1^ is a rocking mode for all molecules, and that at 1049 cm^−1^ is a twisting mode for all molecules. All of the peaks between these frequencies are mixtures of the three vibration types. This indicates that the rocking mode possesses the lowest energy while the twisting vibration has the highest. Three main peaks of Raman scattering are seen in Figure 2 at 648 cm^−1^, 737 cm^−1^, and 968 cm^−1^. The analysis of vibration modes for these three peaks shows that they are all mixtures of rocking, wagging, and twisting vibrations.

There are 34 normal modes in the bending zone, with a range from 1643 cm^−1^ to 1709 cm^−1^. In our previous studies, it was found that the vibrational energy increases from in-phase bending to out-of-phase bending. Due to the disordered structure of hydrogen, the synchronous opening and closing of all molecules is impossible. Figure 5 illustrates the minimum frequency of bending at 1643 cm^−1^. In this vibration mode the numbers of in-phase and out-of-phase motions are almost identical. The greatest frequency is at 1709 cm^−1^ and is an equal mixture of the two bending mode types. The Raman spectrum shows only one peak at 1682 cm^−1^, and the intensities of both Raman and IR are almost undetectably weak. However, peaks in this region can be detected in the INS experiment, as shown in the PDOS spectrum. 

In the intramolecular O–H stretching region, each molecule has two kinds of vibration mode: symmetric stretching and asymmetric stretching. There are 68 normal modes in the stretching zone, with a range from 3105 cm^−1^ to 3368 cm^−1^. In Figure 5, four representative modes are selected, with the typical vibrations of the molecule shown in gold. To maintain a static center of mass, the vibrations of molecules are correlated in a crystal lattice. The minimum-energy mode at 3105 cm^−1^ is predominantly symmetric stretching with some coupled asymmetric stretching. Four peaks were detected by Raman scattering as shown in Figure 2. The minimum-energy peak is at 3180 cm^−1^ and is still mainly symmetric stretching. The highest-energy peak in the Raman spectrum in this region is at 3236 cm^−1^, in which most molecules undergo asymmetric stretching. The highest-energy peak of the entire Raman spectrum is at 3363 cm^−1^, in which almost all of the vibrations are asymmetric. The increasing trend of vibrational energies in this region from symmetric to asymmetric stretching is consistent with the literature [28]. Owing to selection rules, the IR absorptions are mostly symmetric stretching. In Figure 5, one can see that the highest-intensity peak of IR is at around 3180 cm^−1^.

## 4. Conclusions

Ice XVI, the first negative-pressure clathrate ice produced in the lab, was simulated using the first-principles DFT method. The vibrational spectra of INS, IR absorption, and Raman scattering were calculated theoretically. Four regions of the spectra are discussed individually together with some typical vibration modes. Compared with calculated results of ice Ih, its density is 0.77 kg/cm^3^, much lower than 0.88 kg/cm^3^ of Ih. Accordingly, the lengths of the H-bonds in ice XVI are bigger than Ih, manifesting an obvious red-shift of the H-bond (for the upmost mode, 314 cm^−1^ vs. 327 cm^−1^), and a slight blue-shift in the stretching region [25]. In the translational region, two H-bond peaks were found similar to the spectrum of ice Ic [27]. These H-bond vibrations could be classified into two groups based on their physical motions. The strong peak originated from four H-bonds of one molecule vibrating together while the weak peak corresponded to two H-bonds vibrating. We confirmed that this is a general rule that also applies to clathrate hydrates.

Based on this study, we also found a rule for the number of vibration modes in the four vibration regions in the ice. Assuming that the number of molecules in one unit cell of ice is N, the total number of optical normal modes should be 3N × 3 − 3. The O–H stretching region contains 2N modes corresponding to the two kinds of molecular stretching: symmetric and asymmetric. The bending region contains N modes as there is only one such mode for each molecule. The librational band has 3N modes representing the twisting, wagging, and rocking vibrations. Finally, the number of optical normal modes in the translational band is 3N − 3. The coefficient is 3 because there are three spatial vibration modes for each molecule, while 3 acoustic modes must be subtracted. This is a general formula governing any ice phase.

## Figures and Tables

**Figure 1 materials-12-00246-f001:**
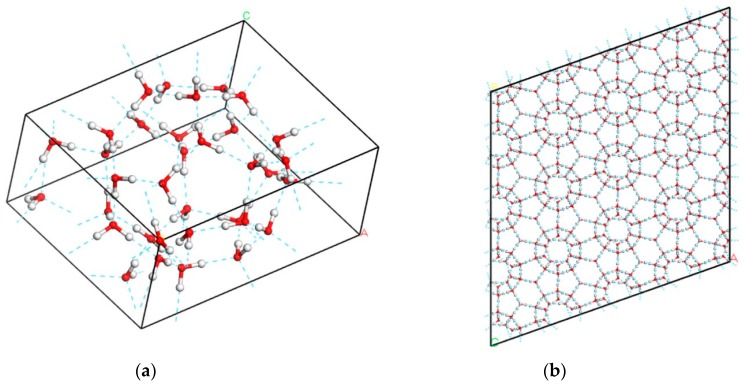
Diagram of ice XVI structure. (**a**) Unit cell of 34 molecules. The red balls indicate the oxygen atoms and the hydrogen atoms are in gray. The hydrogen bonds are shown with blue dashed lines; (**b**) Side view of a 4 × 4 × 4 supercell.

**Figure 2 materials-12-00246-f002:**
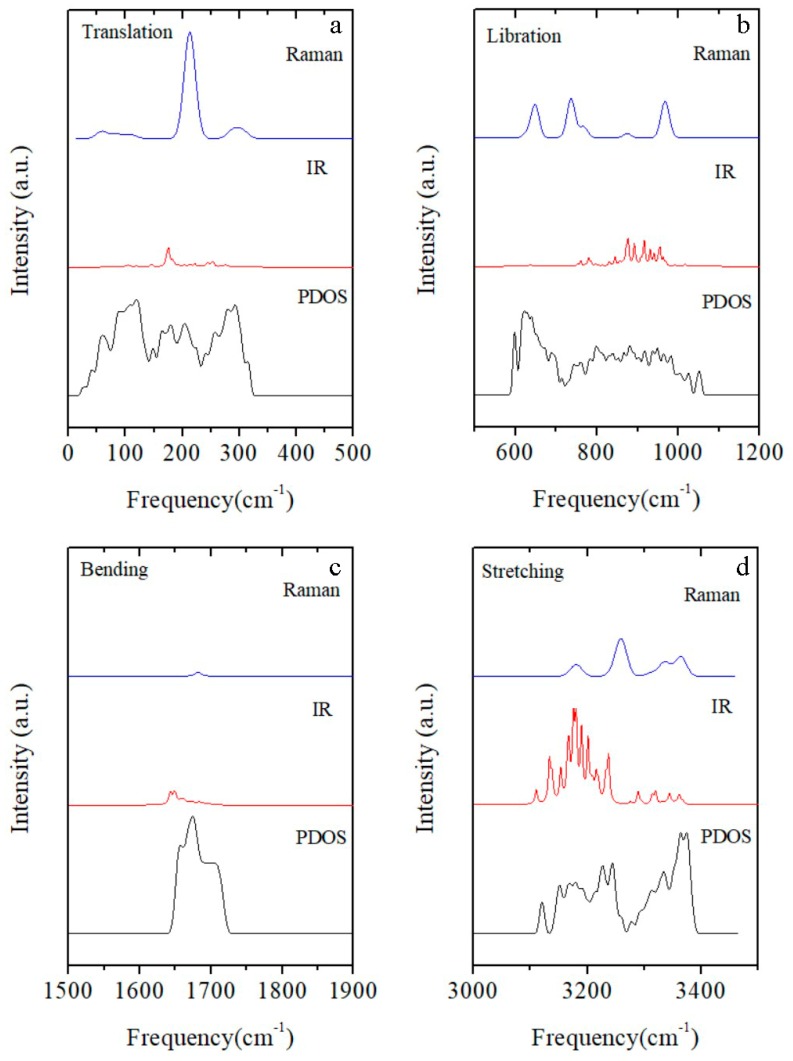
Vibration spectra of ice XVI simulated with the CASTEP code. From top to bottom: Raman scattering, IR absorption, and PDOS, which is proportional to inelastic neutron scattering. (**a**) Intermolecular translation band; (**b**) Intermolecular libration band; (**c**) Intramolecular bending band; (**d**) Intramolecular stretching band.

**Figure 3 materials-12-00246-f003:**
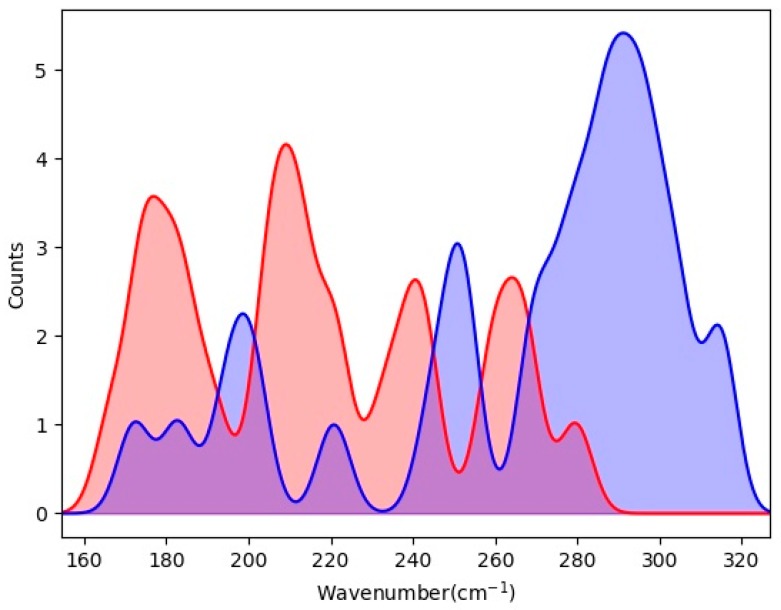
The distribution diagrams of four-bond modes (blue) and two-bond modes (red) of ice XVI in the range from 160 cm^−1^ to 320 cm^−1^. The standard classification of stretching modes in terms of the vibrational direction is applied.

**Figure 4 materials-12-00246-f004:**
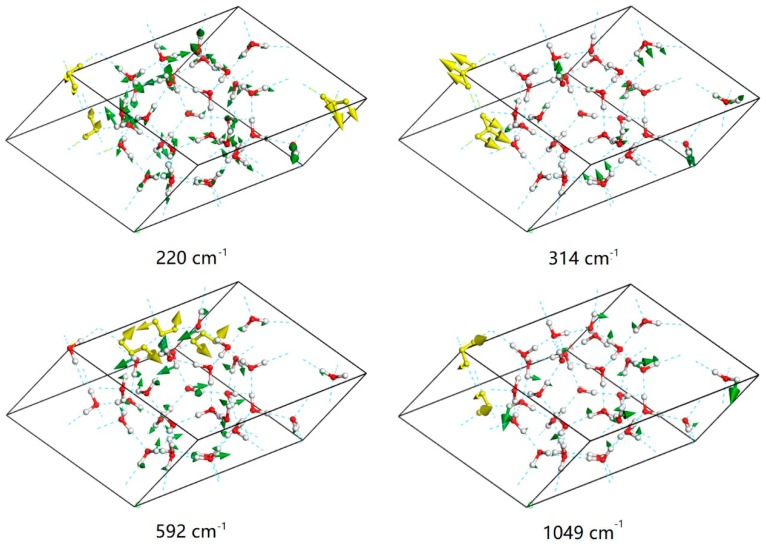
Examples of normal modes of intermolecular vibrations. Representative vibrating molecules are shown in gold. For H-bond vibrations, the molecules present rocking or wagging vibration modes at 220 cm^−1^, while the molecules vibrate along the bisector of the angle for the mode at 314 cm^−1^. The mode at 592 cm^−1^ is the minimum-energy mode in the librational band and involves rocking of all of the molecules. The mode at 1049 cm^−1^ is the maximum-energy mode in this region and involves twisting vibrations for all molecules.

**Figure 5 materials-12-00246-f005:**
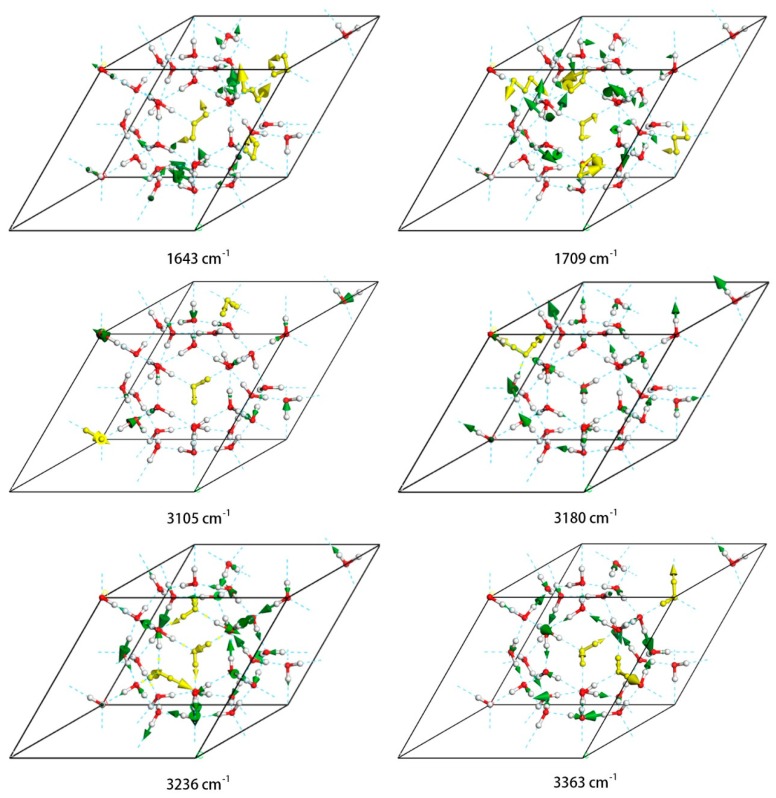
Four examples of normal vibration modes in the intramolecular O–H stretching region (3105 cm^−1^, 3180 cm^−1^, 3236 cm^−1^, 3363 cm^−1^) and two examples for bending (1643 cm^−1^, 1709 cm^−1^). The green arrows indicate the direction of vibration and their thickness is proportional to the amplitude. Typical vibration modes are shown in gold.

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
