# Peer review of "Investigations of the Hydrogen Bonds and Vibrational Spectra of Clathrate Ice XVI"

_materials, 2019, doi:10.3390/ma12020246_

Round 1

Reviewer 1 Report

See attached file.

Author Response

Manuscript ID: materials-419459

TITLE: " Investigations of the hydrogen bonds and vibrational spectrum of clathrate ice XVI "

Our response to the reviewer’s comments:

The authors would like to appreciate the reviewer’s professional comments. Here are the replies to the questions:

1. It is stated several times in the paper that the authors use a 34-molecule supercell. However, on page 2 line 51, it says that “a cage composed of 28 molecules”. Please make sure this a correct description and if so, how does this “28” corroborate with the “34” mentioned elsewhere?

Response:

Ice XVI is S-II type clathrate ice which may traps methane inside due to it has cage structure. A supercell possesses one cage. And a cage composed of 28 molecules. Please see the video for details.

2. On page 2 line 62, the “(depicted below)” is confusing. Please specify which figure is related to this sentence.

Response:

The phrase means figure 3. It was deleted due to confusion here.

3. Page 2 line 65, the sentence “the molecule possessing the largest amplitude in each mode” is not clear and needs correction.

Response:

Thank you, this sentence was corrected as below:

To clearly characterize the complex vibration modes, which are a key property of ice hydrates, we distinguished the mode types by one molecule in supercell which possessing the largest amplitude.

4. Page 6 line 149, it is said that “peaks in this region were detected from the INS experiment”. If true, please provide a reference to support this claim. 

Response:

Sorry, this is a misnomer. No literature presented INS till now. The sentence was revised as below:

However, peaks in this region can be detected in the INS experiment, as shown in the PDOS spectrum.

5. Page 7 line 162, the “consistent with the literature” claim also needs to refer to a corresponding reference.

Response:

Sorry, reference was added.

… symmetric to asymmetric stretching is consistent with the literature [28].

6. Last but not least, the vibrational modes from all 4 regions have been described in the manuscript. However, the comparison of ice XVI to other ice forms is not adequate. It is mentioned several times that those vibrational modes are “similar” to normal ice, but, more importantly, that which mode is not “similar” has not been fully discussed. And which of those vibrational results can be related to the special properties of ice XVI needs to be added to the discussion as well. 

Response:

Since the normal modes are calculated by different supercells under harmonic approximation, we do not compare the mode individually. However, we can discuss the trends for comparison. We inserted some comments in conclusions (line 170):

Four regions of the spectra are discussed individually together with some typical vibration modes. Comparing with calculated results of ice Ih, its density is 0.77 kg/cm3, much lower than 0.88 kg/cm3 of Ih. Accordingly, the lengths of the H-bonds in ice XVI are bigger than Ih, manifesting an obvious red-shift of H-bond (for the upmost mode, 314 cm-1 vs. 327 cm-1), and a slight blue-shift in the stretching region [29]. In the translational region, two H-bond peaks were found, similarly to the spectrum of ice Ic [31].

Reviewer 2 Report

The manuscript by Wang and co-workers presents a very interesting analysis of the theoretically estimated vibrational modes of an uncommon phase of ice, namely, ice XVI. It continues the tradition of the authors in estimating inelastic neutron scattering, Raman scattering and IR absorption of ices and related materials. In my opinion, the calculations were suitably performed and the data analyzed properly. I also judge the manuscript very concise and clear. Therefore, it fulfills the criteria for publication in Materials and would be of interest to the journal readership. It should be accepted for publication after very few minor improvements, which I specify in the following:

Line 38. Here it would be useful to mention the energy cut-off and the number of K-points previously used to perform the energy calculation.

Line 45. The meaning of the sentence “… locations of 50% possibility were labelled.” is unclear. Please, clarify it.

Line 54. Here it is not clear which hydrogen-bond property is underestimated by the RPBE functional. In case it underestimates hydrogen-bond distances only, the theoretical crystal density should be higher than the experimental one, which is not the case. Please, clarify this point.

Line 173. I think that the 3N-3 rule was not found firstly in this work, as it was largely used in previous studies of the same authors. Better rather emphasize it was confirmed.

Author Response

Manuscript ID: materials-419459

TITLE: "Investigations of the hydrogen bonds and vibrational spectrum of clathrate ice XVI"

Our response to the reviewer’s comments:

The authors would like to appreciate the reviewer’s professional comments. Here are the replies to the questions:

1. Line 38. Here it would be useful to mention the energy cut-off and the number of K-points previously used to perform the energy calculation.

Response:

We inserted a sentence as below (in red) to account for the energy cut-off and K-point, please see the revised manuscript in line 43.

The energy cut-off was set as 750 eV and the K-point separation was 0.07/Å. Norm-conserving pseudopotentials

2) Line 45. The meaning of the sentence “… locations of 50% possibility were labelled.” Is unclear. Please, clarify it.

Response:

The hydrogen atoms in crystal are disordered. Each hydrogen atom has two possible positions along the O…O chain, and each position has a 50% probability. We deleted this sentence due to its ambiguous.

3) Line 54. Here it is not clear which hydrogen-bond property is underestimated by the RPBE functional. In case it underestimates hydrogen-bond distances only, the theoretical crystal density should be higher than the experimental one, which is not the case. Please, clarify this point.

Response:

The RPBE function slightly underestimate the H-bond force. Thus the length of H-bond increases. That’s why the density is lower than experimental data.

4) Line 173. I think that the 3N-3 rule was not found firstly in this work, as it was largely used in previous studies of the same authors. Better rather emphasize it was confirmed.

Response:

Thank you very much for your elaborate review. According to solid state theory, to calculate the lattice vibrating phonons, there are 3N-3 optic branches under Harmonic approximation. We thus obtained 3N-3 normal modes. N means atom numbers in one primitive cell. In this study, N means one molecules. So the normal modes are 3N × 3 – 3. For the translational region, there are 3N-3 normal modes. This rule was found by the first author, Ze-Ren Wang, and maybe cited in our other papers.
